# Human Leucocyte Antigen System and Selection of Unrelated Hematopoietic Stem Cell Donors: Impact of Patient–Donor (Mis)matching and New Challenges with the Current Technologies

**DOI:** 10.3390/jcm12020646

**Published:** 2023-01-13

**Authors:** Roberto Crocchiolo, Gianni Rombolà

**Affiliations:** 1Servizio di Immunoematologia e Medicina Trasfusionale, ASST Grande Ospedale Metropolitano Niguarda, Piazza dell’Ospedale Maggiore, 3, 20162 Milano, Italy; 2Laboratory of Immunogenetics and Transplant Immunology, Azienda Ospedaliero-Universitaria Careggi, 50134 Firenze, Italy

**Keywords:** HLA, unrelated donors, stem cell transplantation, next-generation sequencing, mismatch

## Abstract

The selection of hematopoietic stem cell donors for allogeneic transplantation (allo-HSCT) is mainly driven by human leucocyte antigen (HLA) matching between patient and donor, with HLA-identical matched siblings being the preferred choice in most situations. Although other clinical and demographical variables matter, especially, donor age, which is unequivocally associated with better transplant outcomes, the histocompatibility criteria have a central role in the search for the best donor, particularly in the setting of unrelated allo-HSCT where HLA disparities between patient and donor are frequent. The present review is focused on the role of HLA incompatibilities on patient outcome according to the most recent literature, in an attempt to guide transplant physicians and search coordinators during the process of adult unrelated-donor selection. The technological progresses in HLA typing, i.e., with next-generation sequencing (NGS), now allow disclosing a growing number of HLA incompatibilities associated with a heterogeneous and sometimes unknown spectrum of clinical severity. Their immunogenic characteristics, i.e., their position inside or outside the antigen recognition domain (ARD), their permissiveness, their intronic or exonic nature and even the expected expression of the HLA loci where those mismatches occur, will be presented and discussed here, integrating the advances in the immunobiology of transplantation with survival and toxicity outcomes reported in the most relevant studies, within the perspective of improving donor selection in the current practice.

## 1. Introduction

The selection of hematopoietic stem cell donors for allogeneic transplantation (allo-HSCT) is mainly driven by human leucocyte antigen (HLA) matching between patient and donor, with HLA-identical matched siblings being the preferred choice in most situations [1]. The increase of unrelated donations for allo-HSCT in last years [2] and the technological progresses in HLA typing [3] entail today a growing number of HLA incompatibilities disclosed between patients and their respective donors, with a heterogeneous and sometimes undefined spectrum of clinical significance. The immunogenic characteristics of the incompatibilities, i.e., their position inside or outside the antigen recognition domain (ARD) or their immunogenicity, the permissiveness with respect to transplant outcomes, their intronic or exonic nature and the expression of the HLA locus where those mismatches occur, are all elements to be taken into account before allo-HSCT is performed, in order to eventually adapt graft-versus-host disease (GvHD) prophylaxis strategies and optimize patient outcome. Nowadays, over 40,000,000 unrelated donors are potentially available from worldwide registries [4], and sources of HLA-mismatched hematopoietic stem cells like cord blood or haploidentical donors are readily available for allo-HSCT; therefore, the donor selection process requires up-to-date knowledge and cooperation between HLA laboratories, search coordinators and transplant physicians. As a consequence of the evolution of the transplantation procedures, HLA typing technologies and immunogenetics progresses, the HLA incompatibilities between patients and donors should be considered today from a renewed perspective, going beyond the “old” definitions of antigenic or allelic, to now include concepts of permissiveness and immunological risk. 

We present here the most recent literature on HLA and allo-HSCT from adult unrelated donors by providing evidence about the immunological and clinical meaning of HLA incompatibilities between patients and their respective donors. Finally, integrating the advances in the immunobiology of transplantation with clinical experience is warranted to sustain and improve patient survival after allo-HSCT. 

## 2. Definitions of HLA Mismatch in allo-HSCT

Except for HLA-identical siblings, all unrelated stem cell donors present some degree of HLA incompatibilities with their patients even in the best HLA-matched situation, due to the fact that the HLA haplotypes are not those inherited from the parents. The comparable clinical outcomes after allo-HSCT from a well-matched unrelated donor and an HLA-identical sibling [5,6,7] are explained by the fact that the above-cited HLA incompatibilities do not involve the ARD of the loci known to affect survival, being classically HLA-A, -B, -C and DRB1 [8] as well as, to some extent, DQB1 [9]. Of course, factors other than HLA do impact patient survival after allo-HSCT, among which, age matters most [10], suggesting the superiority of a younger well-matched unrelated donor upon an older HLA-identical sibling [11].

Beyond the cited HLA loci, DPB1 matching has been extensively studied in allo-HSCT, and some combinations (referred to as permissive) demonstrated to be better tolerated than others, based on the immunogenicity of the associated T-cell epitopes [12,13,14] and HLA expression [15,16]. Due to weak linkage disequilibrium, indeed, DPB1 is frequently mismatched in 10/10 HLA-matched patient–donor pairs [17]; therefore, matching at DPB1 (12/12 HLA-matched) is achievable in a lower proportion of pairs with respect to 10/10 HLA matching and permissiveness [17].

Noteworthy, the recent introduction of second-generation sequencing technologies for routine HLA typing (next-generation sequencing, NGS) has allowed for the detection of incompatibilities occurring beyond the second field [3], whose clinical significance is less defined than that of incompatibilities within the ARD, albeit under investigation. Recently, improved survival was observed among patients receiving allo-HSCT from donors matched at ultra-high resolution [18]; nonetheless, the current standards contemplate the first two fields for clinical significance of HLA mismatches [19]. Future refinements cannot be excluded, as growing data will allow for further association analyses. A recent national study [20] found lower acute GvHD when patient and donor shared the same geographical origin, possibly related to higher HLA similarity beyond the second field. 

The so-called “non-classical” HLA loci (i.e., HLA-G and -E) might also have a clinical impact [21,22], although this is beyond the scope of the present article. 

All these elements together demonstrate that the definition of clinically meaningful HLA incompatibilities in allo-HSCT are not fixed but change over time according to the technological advances in HLA typing coupled to clinical studies. 

Based on the most prominent current literature and respective evidence, we will focus here on the clinical relevance of HLA incompatibilities within the ARD. The effect of such incompatibilities on the physico-chemical structure of the HLA molecules and their bound peptides (immunopeptidome), the direction of mismatches and the expression of the loci involved have to be taken into account for a proper interpretation [19] and to finally deal with permissiveness and immunological risk. 

## 3. Methodology of Most HLA Studies in the HSCT Setting

The most relevant studies reporting the clinical impact of HLA matching between transplanted patients and their adult unrelated donors are retrospective, mono- or multicenter and even registry-based, using univariate and multivariate regression analyses for any association between HLA matching and clinical outcomes. As a consequence, the evidence generated is limited due to the lack of prospective interventional trials that are unfeasible in this setting. Nevertheless, the current recommendations and up-to-date daily practice among search coordinators and transplant physicians refer to such level of evidence, particularly appreciated when huge numbers of patient–donor pairs are reported, as it occurs in registry studies. Besides this, two meta-analyses have been published so far to our knowledge [23,24], showing similar results. 

The immunological risk associated with the presence of any HLA incompatibility must be firstly assessed by the morbidity and mortality outcomes, mostly, acute GvHD and overall survival, although secondary outcomes such as progression-free survival, non-relapse mortality, relapse incidence, chronic GvHD or even composite endpoints including GvHD-free, relapse-free survival are also reported but with potential discrepancies among studies, due to differences in the definitions of the endpoints or due to reporting bias especially in multicenter or registry analyses. Despite the fact that relapse is a problematic issue after allo-HSCT for malignancies, there is no current evidence that any HLA mismatch confers a reduction of the relapse risk without increasing non-relapse mortality, consistently translating into a lack of survival benefit among the published studies. In addition, relapse and non-relapse mortality are competing risks, meaning that patients with one event are no longer at risk for the competing one [25]. 

Acute GvHD, especially of grade II or higher [26], that is, when systemic immunosuppressive treatment is needed, is probably the most suitable marker of morbidity among the above-cited studies, mainly because of its easily reproducible evaluation and definition, providing low expected discrepancies between the reported studies; moreover, any reported additional acute GvHD risk associated with any HLA mismatch might be the basis for a reinforced GvHD prophylaxis in the presence of such incompatibility [27]. Secondly, the multivariate hazard ratio together with the 95% confidence interval (CI) provides an estimate of the risk carried on by any mismatch(es) when compared with matched pairs, thus giving valuable information about the respective clinical importance. Noteworthy, the amplitude of the 95% CI often reflects the number of pairs analyzed, an important factor to be taken into account when interpreting the reported results.

While the immunological risk carried on by HLA incompatibilities between patients and their respective donors might be quantified, the concept of permissiveness is more related to the extent transplant physicians and patients accept for the immunological risk in the presence of any HLA mismatch. Of course, this risk acceptance depends on several factors, including disease prognosis, patient wishes and available therapies. Nevertheless, in most current literature, the permissive HLA mismatches usually refer to incompatibilities that do not significantly impair the transplant outcomes [13,28,29].

## 4. Locus-Specific HLA Incompatibilities

Several studies reported significant associations between the HLA matching of patient–donor pairs and transplant outcomes, and the most relevant among them provided the basis for the unrelated donor search and selection over time [8,13,30,31,32,33]. However, as transplantation practices change and evolve [34], many studies reappraised the question in recent years and provided updated results in an allo-HSCT setting where a predominant proportion of peripheral blood stem cell grafts vs. bone marrow, less toxic conditioning regimens, improved GvHD prophylaxis and higher median patient age is generally observed when compared to the picture of the previous studies published in the first decade of the years 2000, referring to allo-HSCTs performed in the late 1980s and 1990s. 

### 4.1. HLA-A

The first among the class I alleles, HLA-A, shows the highest expression, together with HLA-B [35]. When dissecting recent studies (published after 2010) with a relevant number of transplants (>1000), the hazard ratio of mortality is between 1.17 and 1.43, and the grade II–IV acute GvHD risk is from 1.18 to 1.34 times higher in HLA-A mismatched vs. matched pairs (Table 1). There is no significant differential impact on the outcome depending on whether the mismatch is antigenic or allelic; thus, they are comparable. This finding is consistent across all classical HLA loci, with the exception of HLA-C (see below). 

### 4.2. HLA-B

As shown in Table 1, the risks of overall mortality and grade II–IV acute GvHD in the presence of one incompatibility at locus B are 1.20–1.52 and 1.11–2.02 times higher, respectively, when compared to those of matched patient–donor pairs. A recent classification using data from the HLA-B Exon 1 (“B-leader”), proposed by Petersdorf and colleagues [41], informs on the acute GvHD risk and may be considered whenever two or more unrelated donors are B-mismatched with a patient and are potentially suitable for selection. 

### 4.3. HLA-C

With the exception of the C*03:03/C*03:04 mismatch [36], all other incompatibilities at this locus represent a significant risk factor for both morbidity and mortality, to a similar extent as those regarding HLA-A and -B, despite the lower surface expression, which is nevertheless heterogeneous across distinct HLA-C allotypes [37]. While the C*03:03/C*03:04 mismatch is not associated with impaired survival and can be considered permissive, both allelic and antigen incompatibilities at locus C are similarly detrimental for transplant outcomes [36] and should be considered equivalent in terms of donor choice. Due to lower surface expression levels, mismatches for the allotypes C*03, C*07 or C*17 might be better tolerated than others [37]. 

### 4.4. HLA-DRB1

Being the most expressed among all class II alleles [42], HLA-DRB1 is rarely mismatched in unrelated allo-HSCT, as shown in early studies that reported a significant negative impact of any mismatch at this locus [43], driving the donor search, still focused on DRB1-matched unrelated donor first. As a consequence, the cohorts of DRB1-mismatched patient–donor pairs are quite small, and evidence on any effect is rather low. Only one study [32] found higher mortality in the presence of a DRB1 mismatch, without increase in acute GvHD (Table 1). According to one [24] of the two previously cited meta-analyses, significant higher mortality is associated with mismatch at DRB1 (HR 1.19, 95% CI: 1.07–1.32) but appears to be better tolerated than class I incompatibilities, these latter being associated with hazard ratios ranging from 1.23 to 1.33 [24]. 

### 4.5. HLA-DRB3/4/5

Less investigated and not routinely typed, the HLA-DRB3/4/5 genes appear to be relevant for transplant outcomes. Indeed, two studies so far demonstrated significant impaired survival [39] and increased acute GvHD [38] when mismatches at these genes are present. A previous collaborative analysis suggested a significant role for the DRB4 locus [44]. Moreover, mismatches at HLA-DRB3/4/5 should be avoided in the presence of multiple other incompatibilities (see below, “Low-expression loci”).

### 4.6. HLA-DQB1

Incompatibilities at HLA-DQB1 seem to be less relevant than the other mismatches [20,31,32], possibly due to either a lower expression vs. that of HLA-DRB1 or few patient–donor pairs that are DQB1-mismatched and DRB1-matched (strong linkage), preventing robust findings. The available data suggest that mismatches at DQB1 might be preferred over those at DRB1 or at class I alleles. 

### 4.7. HLA-DPB1 

Consistent evidence, accumulated over years and by different groups, supports the inclusion of DPB1 in the selection process [45], taking into account the permissiveness of mismatches at DPB1 according to the T-cell epitope or the expression model [46]. The last CIBMTR guidelines [19] include the algorithm based on the T-cell epitope model in order to avoid a non-permissive unrelated donor, currently avoidable in at least 71% of cases [47], although a higher fraction is expected whenever HLA-DPB1 is typed upfront for all potential unrelated donors, as it is the case with NGS typing. Notably, although the benefit of avoiding a non-permissive donor is less pronounced for one-locus-mismatched allo-HSCT [14], this selection is still recommended when two or more 9/10 (or 7/8) HLA-matched donors are potentially suitable in the absence of a 10/10 (8/8) HLA-matched one [19]. 

### 4.8. Low-Expression Loci (DQ, DRB3/4/5, DP)

In addition to studies on the “highly expressed” HLA-A, -B, -C and -DRB1 loci, a CIBMTR study addressed the question of whether cumulative mismatches at the “low-expression” loci DRB3/4/5, DQB1 and DPB1 matter [40]. Interestingly, higher morbidity and mortality risks are associated with three or more DRB3/4/5, DQB1 or DPB1 incompatibilities if one A, B, C or DRB1 incompatibility is present (7/8 HLA-matched donor). In the 8/8 HLA-matched group, mismatches at the low-expression loci were not associated with any adverse outcome [40]. 

## 5. Directionality

The vector of an HLA mismatch is bidirectional unless the patient or the donor are homozygous for the mismatched locus; then, host-versus-graft (HvG) or graft-versus-host (GvH) directions characterize the mismatch, respectively. A single study, conducted on myeloablative allo-HSCTs [48], found that unidirectional GvH vector mismatches are comparable to bidirectional ones in terms of clinical risk, which is higher than for unidirectional HvG, thus suggesting that for an HLA homozygous patient, a donor with a mismatch at this homozygous locus (HvG vector) is preferable over a donor with a mismatch at other loci [48]. 

## 6. HLA Allele Combinations

Other approaches have been applied in the allo-HSCT setting in the attempt of identifying combinations of HLA mismatches having different immunogenic implications and thus hypothetically carrying distinct immunological risks [49,50,51,52,53]. These approaches, mainly based on the characteristics of the epitope mismatches or on the similarity of the haplotypes from both individuals, stand on the assumption that the more the HLA haplotypes are similar (in both evolutionary and immunological terms), the less alloreactivity is expected after allo-HSCT, finally resulting in better prognosis [54]. Such models would be intended to be more useful in supporting the unrelated donor selection as the number of HLA incompatibilities increases, as it occurs for organ transplantation that is the setting where most models come from. However, none of them has been included in the current search recommendations to date. Very recently, HLA evolutionary divergence, a measure of haplotype difference based on evolutionary distance, was shown to predict the prognosis of pediatric and young adult patients undergoing allo-HSCT from matched and mismatched unrelated donors [55], strengthening the concept that not all HLA mismatches are equals and opening to innovative ways to look at HLA incompatibilities.

## 7. Mismatches Outside the ARD

NGS allelic resolution typing provides the full nucleotide sequence of all exons and, for class I genes, noncoding regions. Incompatibilities outside the ARD could potentially affect the allele expression (clinically relevant null alleles) and the indirect presentation of HLA-derived peptides; moreover, the regions outside the ARD are informative for haplotype assignment. A report from the National Marrow Donor Program network showed that diversities (including null alleles) outside the ARD are not frequent in fully matched donor–recipient pairs [56]. The impact of these incompatibilities is now under investigation. Another collaborative study found that DRB1*14:01 vs. *14:54 and DRB3*02:01 vs. *02:02 (two common pairs of alleles belonging to the same P-group) mismatches do not affect a transplant outcome [57]. Similarly, the B*44:02 vs. B*44:27 (B*44:02P) mismatch does not induce alloreactivity in vitro and could be permissive [58]. In a retrospective study on 10/10 matched unrelated transplants, fully matched pairs (i.e., 12/12) confirmed by ultra-high resolution (a technique which allows long-range extensive typing) showed a better survival with respect to pairs with any degree of mismatch at this level of resolution [18]. Most of the discrepant findings referred to mismatches in ARD previously not detected, and only 13 out of 810 carried diversity only in non-ARD exons (*n* = 3) and introns (*n* = 10) [59]. A validation multicenter study confirmed the detrimental impact of incompatibilities outside the ARD only on acute GvHD [60]. Taken together, all these data support the principle that the donor selection process should focus only on the sequence of ARD (match at P-group level) and include the common and well-defined null alleles [61].

## 8. HLA and Disease Relapse 

Currently, there is no consistent evidence that HLA incompatibilities confer a reduction of the relapse risk without increasing non-relapse mortality; therefore, it is not routinely recommended to prefer a mismatched unrelated donor over a matched one in an attempt to enhance the graft-versus-malignancy effect, which instead has shown to be elicited by minor histocompatibility antigens [62]. However, a timely available donor, even if HLA-mismatched, may be a suitable choice for patients with high-risk disease and who are in need of urgent allo-HSCT. The impact of HLA mismatch on the outcome, indeed, is less pronounced when the disease risk is higher [63]. 

Conversely, an HLA mismatch between patient and donor is expected to provide a clinical benefit when KIR alloreactivity is present; therefore, it can be exploited as an adoptive immunotherapeutic strategy both in the unrelated and in the haploidentical setting [64,65]. 

It is worth mentioning the phenomenon of “HLA loss” in disease relapse after allo-HSCT from family haploidentical donors [66], leading to the selection of a new donor carrying a distinct HLA haplotype for further transplantation in order to provide post-transplant immunological pressure against those relapsed leukemic clones no longer expressing or downregulating the surface HLA molecules [67]. The downregulation of surface HLA proteins may also occur in relapsed leukemic cells after unrelated allo-HSCT [68,69].

## 9. Translation into Daily Practice 

A pragmatic interpretation of the data from the reported studies allows for their application to the adult unrelated donors selection, a process requiring up-to-date knowledge and a tight cooperation between HLA laboratories, search coordinators and transplant physicians. At present, if multiple 10/10 HLA-matched unrelated donors are potentially available during the search, the donor selection should take into account the DPB1 and DRB3/4/5 loci. Whenever multiple 9/10 HLA-matched unrelated donors are potentially available without any 10/10 HLA-matched one, the selection process should aim at choosing those donors with the less detrimental incompatibilities with respect to clinical outcomes. These principles are summarized in Table 2. 

In the setting of cord blood transplantation, the recommended HLA matching between patient and donor(s) is less stringent, and the number of HLA alleles usually typed is lower, as well as the respective resolution of the allowed incompatibilities. Unit–unit HLA match is not required for double cord transplantation [19].

Given the rapid advances in HLA typing and the major improvements in GvHD prophylaxis, as observed with high-dose post-transplant cyclophosphamide [70,71], for example, the current indications will probably evolve in the next future, driven by the impressive amount of HLA data provided by NGS and the constant increase of allo-HSCTs from unrelated and HLA-mismatched donors. The experience acquired with high-dose post-transplant cyclophosphamide for haploidentical transplants has prompted transplant physicians to enhance its administration in the HLA-mismatched unrelated setting in recent years [72,73], under the hypothesis that such GvHD prophylaxis would be superior to the “conventional” one based on calcineurin inhibitors, in the presence of one or more HLA incompatibilities. Similarly, considerations about the intensity of the conditioning regimen are warranted, since the expected organ toxicity is tightly linked to the risk of acute GvHD, acting as a potential trigger, enhanced by HLA incompatibilities. 

Beyond HLA, the donor age has proven to be the most relevant factor. Although, to our knowledge, there is no consistent study suggesting a better outcome with younger HLA-mismatched donors over older well-matched ones, patient survival after allo-HSCTs has been reported to be higher with young unrelated donors than with older HLA-identical siblings [11]. As a consequence, donor age should be taken into great consideration during the search process, due to its robust and well-documented impact on patient outcome. 

Finally, besides the fact that the technological improvements are expected to allow for NGS allelic resolution typing of patients and donors in all settings, including allo-HSCT from relative and cord blood units, it is worth noting that its use is particularly helpful during the search of adult unrelated donors, where all the above-cited considerations apply and where the large amount of HLA data obtained provide the basis for a potential improvement of patient prognosis. 

## 10. Conclusions

The HLA system has proven to be crucial for the success of allo-HSCT; therefore, an appropriate donor selection process based on up-to-date knowledge and the cooperation between HLA laboratories, search coordinators and transplant physicians is of paramount importance. Here, we provided the most relevant evidence on the role of HLA incompatibilities according to the recent literature in the adult unrelated donor setting, in an attempt to guide all the involved stakeholders during the unrelated donor selection. The technological progresses in HLA typing together with the advances in the immunobiology of transplantation and the significant improvements in GvHD prophylaxis will probably lead this field to a rapidly evolution in the next future, with new and exciting perspectives on the horizon. 

## Figures and Tables

**Table 1 jcm-12-00646-t001:** Clinical impact of HLA matching in the allo-HSCT setting as reported in most recent relevant studies.

HLA Locus	Mismatch	Article	n	Hazard Ratio (95% CI)	
				OS	GvHD II-IV
A	mismatched vs. matched	Woolfrey 2011 [33]	1933	1.17 (0.93–1.47)	1.18 (0.94–1.50)
		Furst 2013 [32]	2646	**1.43 (1.19–1.72)**	ND
		Pidala 2014 [30]	8003	**1.3 (1.2–1.5)**	**1.3 (1.2–1.5)**
		Morishima 2015 [31]	7898	**1.29 (1.17–1.42)**	**1.18 (1.06–1.32)**
		Picardi 2021 [20]	1788	ND (*p* = n.s.)	**1.34 (1.04–1.74)**
		Kekre 2016 [23]	13,446	**1.48 (1.19–1.86)**	ND
B	mismatched vs. matched	Woolfrey 2011 [33]	1933	1.22 (0.90–1.67)	1.11 (0.81–1.52)
		Furst 2013 [32]	2646	**1.52 (1.20–1.93)**	ND
		Pidala 2014 [30]	8003	**1.2 (1.0–1.4)**	**1.3 (1.1–1.6)**
		Morishima 2015 [31]	7898	**1.27 (1.11–1.45)**	**1.28 (1.11–1.48)**
		Picardi 2021 [20]	1788	ND (*p* = n.s.)	**2.02 (1.53–2.67)**
		Kekre 2016 [23]	13,446	**1.45 (1.20–1.75)**	ND
	B-leader mismatched: MT vs. TT	Petersdorf 2020 [16]	17,100	ND	**1.11 (1.04–1.19)**
	B-leader mismatched: MM vs. TT	Petersdorf 2020 [16]	17,100	ND	1.11 (1.00–1.24)
C	mismatched vs. matched	Woolfrey 2011 [33]	1933	**1.41 (1.16–1.70) ^#^**	1.12 (0.90–1.39)
		Furst 2013 [32]	2646	**1.35 (1.17–1.56)**	ND
		Pidala 2014 [30]	8003	**1.3 (1.2–1.5)**	1.1 (1.0–1.3)
		Morishima 2015 [31]	7898	**1.21 (1.13–1.30)**	**1.27 (1.17–1.37)**
		Picardi 2021 [20]	1788	ND (*p* = n.s.)	ND (*p* = n.s.)
		Kekre 2016 [23]	13,446	**1.58 (1.23–2.01)**	ND
	C03:03/C03:04 mismatch	Fernandez-Vina 2014 [36]	7349	0.98 (0.78–1.23)	0.97 (0.76–1.25)
	Ag vs. All mismatch aT C*03 or C*07	Petersdorf 2014 [37]	1975	ND	1.07 (0.75–1.53)
	residue 116 mismatch vs. match	Petersdorf 2014 [37]	1975	1.12 (1.00–1.26)	1.14 (0.91–1.41)
	residue 77/80 mismatch vs. match	Petersdorf 2014 [37]	1975	**1.13 (1.01–1.26)**	1.02 (0.82–1.27)
DRB1		Woolfrey 2011 [33]	1933	1.30 (0.87–1.94)	**1.60 (1.06–1.80)**
		Furst 2013 [32]	2646	**1.42 (1.10–1.82)**	ND
		Pidala 2014 [30]	8003	1.1 (0.9–1.3)	1.2 (0.9–1.5)
		Morishima 2015 [31]	7898	1.09 (0.98–1.21)	**1.24 (1.11–1.39)**
		Kekre 2016 [23]	13,446	1.16 (0.84–1.59)	ND
DRB3/4/5	mismatched vs. matched	Ducreux 2018 [38]	1975	1.10 (0.91–1.33)	**1.43 (1.07–1.90)**
DRB3/4/5	mismatched vs. matched	Tsamadou 2021 [39]	3410	**1.25 (1.02–1.54)**	1.16 (0.89–1.52)
DQB1	mismatched vs. matched	Furst 2013 [32]	2646	1.23 (1.00–1.51)	ND
		Morishima 2015 [31]	7898	1.08 (0.97–1.19)	1.09 (0.98–1.22)
		Picardi 2021 [20]	1788	ND (*p* = n.s.)	ND (*p* = n.s.)
		Kekre 2016 [23]	13,446	0.95 (0.74–1.21)	ND
DPB1	single-mismatched vs. matched	Pidala 2014 [30]	8003	ND	**1.4 (1.2–1.6)**
	double-mismatched vs. matched	Pidala 2014 [30]	8003	ND	**1.6 (1.3–1.9)**
	matched vs. TCE permissive	Fleischhauer 2012 [14]	8539	0.96 (0.87–1.06)	ND
	TCE nonpermissive vs. permissive	Fleischhauer 2012 [14]	8539	**1.15 (1.05–1.25)**	ND
	TCE nonpermissive vs. permissive	Pidala 2014 [30]	8003	**1.2 (1.1–1.4)**	1.1 (1.0–1.3)
LEL (DRB3/4/5, DQ, DP)	≥3 vs. 0 among 7/8	Fernandez-Vina 2013 [40]	3853	**1.45 (1.06–1.96)**	ND
	≥3 vs. 1 among 7/8	Fernandez-Vina 2013 [40]	3853	**1.43 (1.09–1.87)**	ND

^#^ antigen mismatch. List of abbreviations: MT: methionine–threonine genotype; MM: methionine–methionine genotype; TT: threonine–threonine genotype; TCE: T-cell epitope; ND: not done; OS: overall survival; GvHD: graft-versus-host disease. Respective *p*-value is statistically significant in bold.

**Table 2 jcm-12-00646-t002:** Suggestions for the selection of unrelated donors according to the HLA matching status.

(a) Multiple 10/10 (or 8/8) HLA-matched donors:
(1) avoid T-cell epitope DPB1 non-permissive donors;
(2) match at DRB3/4/5 if possible.
(b) Multiple single-locus mismatched (9/10 HLA-matched, or 7/8) donors:
(1) avoid T-cell epitope DPB1 non-permissive donors;
(2) prefer DQB1 (not applicable for 8/8) or C*03:03/C*03:04 mismatches if possible, otherwise:
(3) select DRB1-mismatched donors if possible, otherwise:
(4) select A-, B- or C-mismatched donors ^#^ (all comparable) if possible;
(5) if 7/8, prefer no more than two low-expression loci mismatches;
(6) if patient homozygous, select the donor mismatched at the homozygous loci (HvG vector).

^#^ other than C*03:03/C*03:04.

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
