# Peer review of "Human Leucocyte Antigen System and Selection of Unrelated Hematopoietic Stem Cell Donors: Impact of Patient–Donor (Mis)matching and New Challenges with the Current Technologies"

_jcm, 2023, doi:10.3390/jcm12020646_

Round 1

Reviewer 1 Report

The authors nicely reviewed the current knowledge on HLA mismatch in the donor selection process, giving a good global overview on this field.

I would suggest expanding the final paragraphs commenting on the effects of HLA mismatch on disease relapse (HLA-loss relapse, etc ...) and how HLA mismatch could be exploit in very high-risk diseases (despite, as acknowledge by the authors, the risk of increasing NRM).

I would suggest commenting on the impact of HLA mismatch in different graft sources (cord blood, etc).

Finally I would also add some comments on the impact of different intensity conditioning regimen and different GVHD platforms (PT-Cyclophosphamide, etc) on HLA-mismatch selection.

Author Response

Reviewer 1

The authors nicely reviewed the current knowledge on HLA mismatch in the donor selection process, giving a good global overview on this field.

I would suggest expanding the final paragraphs commenting on the effects of HLA mismatch on disease relapse (HLA-loss relapse, etc ...) and how HLA mismatch could be exploit in very high-risk diseases (despite, as acknowledge by the authors, the risk of increasing NRM).

Thanks for this comment. We added a new paragraph entitled “HLA and disease relapse” in which we commented on the phenomenon of HLA loss, as suggested, and on the potential exploitation of HLA mismatch to prevent relapse through KIR alloreactivity. We modified the reference list accordingly.

I would suggest commenting on the impact of HLA mismatch in different graft sources (cord blood, etc).

As suggested, we added a statement in the paragraph “Translation into daily practice”. In addition, we added in the abstract and in the text the wording “adult unrelated” in order to better specify the focus of the review.

Finally I would also add some comments on the impact of different intensity conditioning regimen and different GVHD platforms (PT-Cyclophosphamide, etc) on HLA-mismatch selection.

Thanks for this comment, we indeed added some comments in the paragraph “Translation into daily practice” together with other statements as requested by the Reviewer 2. We modified the reference list accordingly.

Reviewer 2 Report

Thank you for organizing a large amount of previous work into this review. I appreciate the opportunity to review this manuscript.

Questions:

- As technology improves, there may come a time when NGS is performed in all cases. With the current knowledge and pricing, in which scenarios do you particularly recommend NGS to help select the best donor?

- You mention that younger age confers benefits to the transplant. Based on current data, are there situations where a younger more-mismatched donor is preferable to a matched or better-matched older donor? This may be outside the scope of this review that focuses on the HLA aspect of selection.

Other notes:

- Page 2, line 61, recommend replacing "Unless the case of" with "Except for".

- In table 1, can you define the abbreviations MT, MM, TT, TCE, ND, OS in case readers are not familiar with them?

Author Response

Reviewer 2

Thank you for organizing a large amount of previous work into this review. I appreciate the opportunity to review this manuscript.

Questions:

- As technology improves, there may come a time when NGS is performed in all cases. With the current knowledge and pricing, in which scenarios do you particularly recommend NGS to help select the best donor?

We added a sentence in the paragraph “Translation into daily practice” in which we address this point and endorse the current use of NGS for the adult unrelated transplantation setting. 

- You mention that younger age confers benefits to the transplant. Based on current data, are there situations where a younger more-mismatched donor is preferable to a matched or better-matched older donor? This may be outside the scope of this review that focuses on the HLA aspect of selection.

Thanks for this comment. To our knowledge, at present there is no consistent study showing a benefit using a younger mismatched over a matched older donor, however the question is very important and we believe that donor age has to be taken into account and must not be underestimated with respect to HLA; for this reason we had cited at the beginning of the manuscript the work from Kroeger et al. on Leukemia 2013 (ref. 11). We also added a sentence in the paragraph “Translation into daily practice” to comment on this question.      

Other notes:

- Page 2, line 61, recommend replacing "Unless the case of" with "Except for". Done

- In table 1, can you define the abbreviations MT, MM, TT, TCE, ND, OS in case readers are not familiar with them?

As requested, we provided the definitions of the abbreviations at the end of the table 1.